# Depth-Based Measurement of Respiratory Volumes: A Review

**DOI:** 10.3390/s22249680

**Published:** 2022-12-10

**Authors:** Felix Wichum, Christian Wiede, Karsten Seidl

**Affiliations:** 1Fraunhofer IMS, 47057 Duisburg, Germany; 2Department of Electronic Components and Circuits, University of Duisburg-Essen, 47047 Duisburg, Germany

**Keywords:** tidal volume, vital capacity, contactless measurement, kinect, structured light, time-of-flight, respiratory measurement

## Abstract

Depth-based plethysmography (DPG) for the measurement of respiratory parameters is a mobile and cost-effective alternative to spirometry and body plethysmography. In addition, natural breathing can be measured without a mouthpiece, and breathing mechanics can be visualized. This paper aims at showing further improvements for DPG by analyzing recent developments regarding the individual components of a DPG measurement. Starting from the advantages and application scenarios, measurement scenarios and recording devices, selection algorithms and location of a region of interest (ROI) on the upper body, signal processing steps, models for error minimization with a reference measurement device, and final evaluation procedures are presented and discussed. It is shown that ROI selection has an impact on signal quality. Adaptive methods and dynamic referencing of body points to select the ROI can allow more accurate placement and thus lead to better signal quality. Multiple different ROIs can be used to assess breathing mechanics and distinguish patient groups. Signal acquisition can be performed quickly using arithmetic calculations and is not inferior to complex 3D reconstruction algorithms. It is shown that linear models provide a good approximation of the signal. However, further dependencies, such as personal characteristics, may lead to non-linear models in the future. Finally, it is pointed out to focus developments with respect to single-camera systems and to focus on independence from an individual calibration in the evaluation.

## 1. Introduction

Respiration is a fundamental function of the human organism and respiratory disorders, such as chronic obstructive pulmonary disease (COPD) or asthma, are among the leading causes of death worldwide [1]. For this reason, the functioning of the lungs is examined in pulmonary function tests. Diseases can thus be diagnosed in time and treated at an early stage. In everyday clinical practice, spirometers and whole-body plethysmographs are used to determine respiratory volumes. However, these methods cannot be used in all application scenarios, for example, because the active cooperation of the patient is required [2] or facial muscle weaknesses prevent the wearing of a necessary mouthpiece [3]. Due to the fact that they counteract these disadvantages, contactless measurement methods are in the focus of research in recent years, especially for mobile applications.

Using depth-based plethysmography (DPG), it is possible to use depth information to infer volume changes in the upper body and thus respiratory parameters. It is possible to determine respiratory motion as well as respiratory volumes and other respiratory parameters. While extensive reviews in recent years have focused on camera-based vital signs monitoring [4], respiration rate measurement [5], non-contact measurement of the respiration rate [6], contactless measurement of respiration in general [7] and DPG measurement in general [8], the main contributions of this paper are: (1) to provide an overview of the state of the art in DPG measurement of respiratory volumes and (2) discuss current developments of DPG with regard to the targeted benefits.

## 2. Fundamentals of Respiratory Measurement

This section serves as an introduction to the general measurement of respiratory parameters. Gold standards are presented and references for further procedures are presented. Respiratory parameters are explained.

The gold standards in the measurement of respiratory volumes are spirometry and body plethysmography with a permissible error of ±2.5% after verification with a 3-L syringe (±0.5% error) [9]. Spirometry measures as a function of time how much volume a person can inhale or exhale. The respective subject breathes through a mouthpiece into a spirometer. Lung volumes can be measured via a change in volume in the system, or in modern devices as an integrated change in flow. The most common techniques involve a turbine that is moved by the respiratory flow or a pneumotachograph, which uses a pressure difference at an artificially inserted resistance to infer the respiratory flow [10]. Whole-body plethysmography, unlike spirometry, can provide information about residual lung volume and total lung capacity. The patient sits in a closed chamber. The opposing variables during breathing can be determined by means of the volume or pressure constancy of the chamber. The measurement method results in various requirements for the stiffness of the chamber, heat and moisture transfer, and calibration [11]. Other methods of measuring respiratory volumes result from computed tomography [12], capnography [13], acoustic monitoring [14], impedance pneumography [15], doppler radar [16], electrocardiography [17], and accelerometers [18].

Important parameters in the clinical evaluation of respiration are respiratory volumes. Figure 1 shows the changes over time in inhaled and exhaled volume. Based on quiet (tidal) breathing, the tidal volume (TV) can be determined. The inspiratory capacity describes the maximum volume that can be inhaled after exhalation. Expiratory vital capacity is defined as the volume between maximum inspiration and maximum expiration. The residual volume always remains in the lungs and the total lung capacity indicates the maximum volume that the lungs can hold. Functional residual capacity describes the volume staying in the lung after tidal breathing [10].

## 3. Review on Depth-Based Respiratory Measurement

This section describes the current published developments of DPG. It is first explained (Section 3.1) how the relevant papers were chosen. Then, the basic methods of DPG are presented (Section 3.2) and the advantages for practical use are derived (Section 3.3). Subsequently, the crucial questions for a real application of DPG are answered: In which measurement scenarios are DPG used (Section 3.4)? Which systems are used for recording (Section 3.5)? Which image area is selected and by which method (Section 3.6)? How is the signal reconstruction performed (Section 3.7)? How accurate are the DPG measurements in the literature and how are they evaluated (Section 3.8)?

### 3.1. Literature Research

Eligible studies were identified through searches of the Scopus and PubMed database, with the corresponding search term: ((depth OR camera OR rgbd OR rgb-d OR kinect OR 3d OR 3-d OR orbecc) AND (lung OR pulmonary OR respiratory OR respiration) AND (volumes OR volume OR tidal AND volume OR vital AND capacity OR volumina)).

Based on the 366 hits on Scopus and 187 hits on Pubmed, 153 duplicates were removed, remaining N = 400. With the inclusion criteria that (1) depth images are evaluated and (2) respiratory parameters are determined; and the exclusion criteria for studies with CT, MRI, or on animals, the found papers were checked for their suitability after reading the title (N = 39) and abstract (N = 20) and based on the results further papers (N = 7) were added manually. A total of 26 papers remained for further evaluation.

### 3.2. Methods of Depth Measurement

Depth information can be provided by stereoscopic camera sensors (SC), structured light sensors (SL), time-of-flight sensors (TOF), or a combination [19]. For DPG, depth information is used to infer respiratory volumes and other respiratory parameters. Based on the sources of this depth information, DPG can be divided as follows:**marker-based methods**This involves placing clearly visible markers on the patient’s upper body, which are then automatically registered by software. The structure of the chest can be calculated via reconstruction procedures after extensive calibration. This marker-based method is also referred to in the literature as opto-electronic plethysmography (OEP) [20] and is shown in Figure 2a. The number of markers is not fixed and can vary from 5 [21] to 89 [22] markers. Marker positioning depends on the area of the thorax being observed and is not limited to one side [23]. Another example of the application is the motion capture system in movies.**direct methods**Using TOF or SL methods, depth information can be inferred without applying markers. TOF measures distance by emitting laser pulses. These pulses are reflected by objects and then picked up again by a detector. Based on the required travel time, the distance can be determined via the speed of light. [24] In the structured light method, on the other hand, a known light pattern is projected onto the scene in the near-infrared range. The distance can be inferred from the deformations of the pattern on surfaces. [25] Stereoscopy is based on the use of multiple, offset cameras. The depth of information can be derived from this offset. The result of direct methods is a point cloud of depth information. The principle of these methods can be seen in Figure 2b.

A further distinction is considered in terms of the number of sensors used, so DPG can be divided into:**single camera systems**A single camera is used to record the subject from one side, mostly frontal.**multi-camera systems**Multiple cameras are used to create a slightly offset stereoscopic effect or to directly view multiple sides of the patient. In particular, an effort is made to create an additional backsight of the patient.

### 3.3. Advantages and Application Scenarios

In contrast to traditional spirometry, DPG has several advantages:**The mechanics and contribution of respiratory motion are made visible**The contributions to respiratory movement by the individual regions of the thorax can be specifically visualized and evaluated. This includes, for example, different respiratory mechanics in persons such as swimmers [26], dancers [27], or infants [28]. In addition, it is conceivable that asynchronous muscle weaknesses can be visualized, or even the failure of a lung lobe. This is not possible with traditional spirometry [29,30]. With respiration rate, respiratory volumes, and chest movements, DPG enables the measurement of three of the four classes of respiratory assessment. Only the concentration of gases cannot be measured with DPG [7].**DPG corresponds to natural breathing**No mouthpiece is needed for non-contact measurement. Such a mouthpiece cannot be used by all patient groups. Especially in the case of facial muscle weakness, deviations in the measurements may occur [31]. Other patient groups, such as with tracheostomy, cannot use such a mouthpiece in the first place [32]. DPG can be performed without active patient participation for tidal volumes, as no mouthpiece is required. A non-contact measurement at rest can be performed straight forward, especially for children, hearing-impaired, learning-impaired, or with language barriers. Thus, breathing is not influenced by further boundary conditions.**DPG is a potential mobile, lightweight, and low-cost method**Apart from the level of development and the technology used, DPG processes offer the possibility to be used easily and everywhere, without the need for trained persons. This is not the case for multi-camera systems that require further calibration or the use of markers that need to be applied for volume extraction. Single camera systems with depth sensors, such as described in [33], which can determine respiratory volumes without calibration, offer the advantages described above. With the proliferation of depth sensors in mobile smartphone cameras [34], such technologies can potentially and in the future enable easy measurement of respiratory parameters in the everyday life of patients. Compared to the whole-body plethysmograph, with potential problems due to claustrophobia [35], DPG is not constrained by spatial constraints and can be used in a mobile manner. Used depth sensors [see Section 3.5] are cheap compared to gold standard technology, furthermore, no further consumables are needed.

The areas of application for DPG are derived from these advantages. In the clinical field, a diagnosis of respiratory diseases such as COPD or asthma is of particular importance. Application scenarios, therefore, extend to the clinical area, as a substitute when other devices cannot be used; for example, due to claustrophobia, or facial muscle weakness; as a mobile application in the home and care sector for spontaneous and mobile monitoring of respiratory parameters without trained personnel; for the evaluation of respiratory mechanics; in areas difficult to access or for non-contact monitoring in, for example, nursing homes, hospitals, or prisons.

While DPG has some advantages, it is not possible to determine all parameters that could be obtained via body plethysmography. This includes for example residual volume and thus total lung capacity, as well as airways resistance.

### 3.4. Settings

Respiratory volumes are assessed via DPG in the standing [36], lying [28], or sitting position [37]. A fixed positioning of at least a part of the body offers the advantage of suppressing arbitrary movements. Especially during strong breathing maneuvers, the upper body may actively support the breathing movement. In this case, the entire upper body moves forward during inhalation and backward during exhalation [38]. These movements are suppressed when sitting or lying, which simultaneously raises the signal-to-noise ratio (SNR) [7]. In addition, it can be distinguished whether backrests [39] or armrests [26] are used or not. An example of such a measurement setup is shown in Figure 3.

Clothing is another parameter influencing SNR [8]. Wrinkling can result in the actual chest movement not being visible [40]. However, strongly reflecting surfaces can likewise weaken the signal in these areas [41]. For this reason, the clothing worn is sometimes limited to tight-fitting clothing [36], while in other systems an unclothed upper body is measured [42]. The latter applies in particular to the placement of markers.

A parameterization study by John et al., in 2016 examines various influences on the determination of respiratory rate with a depth camera. It is shown that generally better results are achieved in a sitting position compared to a lying position. This is due to body movements that are superposed on respiratory movements. A (partial) covering leads to the highest errors. A distance of 1–2 m is determined as the optimum distance. A higher distance reduces the signal quality with a lower SNR. For coverage, on the other hand, a higher distance can offer advantages, which is explained by lower shadowing and reflection effects [41].

The respiratory parameter measurement procedure should follow the American Thoracic Society and European Respiratory Society guidelines for clinical evaluation [9,10].

### 3.5. Recording Systems

The sensors used for DPG recording are standard RGB cameras, or even smartphone cameras, TOF, or SL sensors. A large comparison of conventional depth cameras from the fields of TOF, SL, and SC was performed in Giancola et al. 2018 [44]. Among other cameras, the state-of-the-art 3D cameras Kinect v2 (TOF), Orbbec Astra (SL), and Intel D435 (SC) were compared. The uncertainty of TOF cameras increases linearly with depth, while the triangulation principle of SL and SC leads to a quadratic increase of uncertainty with higher distance. For this reason, SL cameras may be preferred for close-range applications, especially without further outdoor environmental influences such as sunlight.

Complete recording systems are available for purchase exclusively for OEP, for example, OEP System (BTS Bioengineering) or Motion Analysis (Santa Rosa, CA, USA). For Kinect cameras, Soleimani et al. 2016 [45] provide an automatic, open-source data acquisition approach. For this purpose, two opposing Kinect v2 cameras are calibrated and the transformation parameters are estimated. These are then used to align the point clouds and register them on a common plane. This method was created specifically for recording and evaluating respiratory volumes.

### 3.6. ROI-Selection

Based on the recorded depth data, an ROI can be selected for further signal processing. The ROI contains relevant areas, such as the upper and lower thorax, that contribute significantly to respiration. The selection of an ROI thus reduces the computational capacity and increases the SNR. In principle, an ROI can be selected automatically, semi-automatically, or manually, remain statically fixed for the whole measurement, or can be changed dynamically. The selected regions will also be discussed in the following.

By applying markers, marker-based methods make a manual selection of an ROI obsolete. The markers and thus elevation and depression of the chest are recorded dynamically [21,22,26,27,32,46,47]. In contrast, an automatic selection of an ROI based on key points is done by skeletonization in Soleimani et al. 2015. A rectangular ROI is placed on the subjects’ chest, depending on the skeletal joints ShoulderRight, ShoulderLeft, SpineShoulder, and SpineMid, derived from the Kinect v2 camera model [37,48]. The manual setting of such a region is used by Harte et al. 2016 [42] or Wiegandt et al. 2021 [28]. The selected ROI is subsequently tracked over the entire time period. Such a procedure allows accurate positioning of the desired ROI, but is not reproducibly repeatable for confirming accurate measurements. The procedure according to Oh et al. 2019 [49] uses spatial and temporal information to define the ROI. The shape of the human chest is used as spatial, a priori information [50], and the segmentation of adjacent time windows as temporal information. A weighting of both pieces of information is used to select regions contributing to respiration, which are defined as outwardly bulging boxes. Another automatic method introduced by Ostadabbas et al. 2016 [40] is based on the mean value image of the entire measurement. Thresholding background pixels and foreground pixels such as the knees with a minimum distance results in an initial reduction of pixels of interest. Contiguous areas are then selected row and column-wise. In the last step, cropping ensures anatomical correctness. A semi-automatic algorithm, the flood fill method, for selecting the ROI is used in [51].

ROIs are used to select regions that contribute to respiration. For this reason, the upper body up to the head is considered. Subregions (SROI) can be a finer subdivision and measure the contribution of different chest regions. SROIs are particularly in use in OEP, where the markers are placed at the boundaries, making it easier to distinguish the regions under consideration. Three SROIs are distinguished in general in OEP measurement [46]: pulmonary rib cage, abdominal rib cage, and the abdomen, as in Figure 4a. Another subdivision is made by Ripka et al. 2014 [21] in the observation of the thorax from a lateral view. Four SROIs are defined via markers: Upper Thorax, Lower Thorax, Upper Abdomen, and Lower Abdomen, see Figure 4b.

The same SROIs, however, from a full-page view are used in Silvatti et al. 2012 [26]. Seppanen et al. 2015 [52] state to use two SROIs, defined as horizontal stripes at the xiphoid process near the umbilicus. With the help of a principal component analysis (PCA), many evenly distributed SROIs are combined into one respiratory signal in Soleimani et al. 2018 [38] to reduce the influence of body movements. Yu et al. 2012 [53] use three SROIs after manually applying a chest model. A distinction is made between the abdomen and the left and right sides of the thorax. After an automatic upper body detection via OpenCV haar cascade classifier [54], a preliminary ROI was set first by Imano et al. 2020 [55]. To determine the final ROI, SROIS are integrated line by line from top to bottom until 90% of the amplitude value of the total ROI amplitude is reached. An example of this algorithm is shown in Figure 5.

### 3.7. Signal Reconstruction

Time signals are derived from the acquired images and the selected regions, respectively. Such a corresponding time signal $S_{Method}$ (with a unit of distance or volume) already correlates strongly with a spirometer signal $V_{Ground\;Truth}$ (in liter), mostly used as ground truth. However, it is necessary to use further processing steps. Finally, the measurement signal must be transformed into the target format $V_{Target}$ (in liter) via a model or a transformation function f, shown in Equation (1).
(1)$$V_{Target}=f\left(S_{Method}\right)$$

The goal is to minimize the error between this target function and $V_{Ground\;Truth}$ by a suitable model and extraction method. Therefore, linear scaling factors are often used. In the following, the individual signal processing steps up to the final measurement signal are considered. Figure 6 shows a flowchart of a whole measurement and training set-up.

Boudarham et al. 2013 [30] determine the measuring signal by using a motion analysis system (Motion Analysis, Santa Rosa, CA, USA). A linear regression function with slope and intercept is used to transform the vital capacities into the target signal. The parameters are collected on the basis of all measurement signals. Another OEP System is used by Feitosa et al. 2019 [22], with BTS Bioengineering (Milanese, Italy) without any further scaling, or with OptiTrack Prime 17W (NaturalPoint, Inc., Corvallis, OR, USA) [27].

Harte et al. 2016 [42] calibrated four cameras for synchronous image recording. Geomagic Studio 2012 software is used to extract a mesh, and afterwards, volumes from the point clouds. The measurement signal is then further processed with cubic interpolation, a Butterworth zero-phase fourth-order bandpass filter, and further down-sampling to 5 Hz. The slope and intercept of a transformation function are determined via total least squares. In the work by Oh et al. 2019 [49], the target signal results from the summation of the differences of successive images. Coefficients describe the relationship between distance and actual pixel length. Ostadabass et al. 2016 [40] use linear scaling between the measurement signal and the target as well. The coefficients result from a reference measurement with ground truth volumes. It is assumed that the scaling coefficients for a given subject are constant. Linear regression, using estimated tidal volume as a regressor and ground truth tidal volume from spirometry as regressand is used by Imano et al. 2020 [55]. Another method for calibrating a linear transformation is used by Reyes et al. 2017 [36]. Indeed, 50% of the test points of each measurement are used for calibration randomly.

The approach of Soleimani and Sharp [37,48,57] is based on two scaling factors, which are applied for the tidal volume on the one hand, but also separately for the vital capacity determination on the other hand. A mesh is generated from the 3D point cloud of the upper body and the enclosed volume is determined with respect to a planar reference surface. Due to the scaling factors, the position of the reference surface is not important. A fourth-order Butterworth filter is used to reduce the over-smooth of a moving average filter. Further filtering with a cutoff frequency of 1 Hz for tidal volume and 3 Hz for vital capacity is applied. Keypoints are extracted from the measurement signal, which is aligned to the local minima and maxima of the measurement signal. In a training phase, the ideal scaling factors are determined using a spirometer signal. In the test phase, the scaling factors are chosen, so that the best-matching scaling factors with the least error are selected from the training data. Training and testing of the scaling factors are performed intra-subject-wise. With the use of a second camera, the measurement signal is calculated as the difference between the measurement signal of the front camera and the measurement signal of the rear camera. Thus, the entire upper body is viewed and artifacts caused by movements are reduced [43]. Later, the authors use PCA for motion artifact reduction. Hereby, a planar surface is projected onto the curved model of the thorax and the volumes of equal-width compartments form the input vectors of the PCA. The principal component then corresponds to a motion-adjusted signal [38]. Empirical Mode Decomposition (EMD) [58] can be used for detrending data [43].

Via a multiple regression test, Ripka et al. 2014 [21] found additional parameters to transform the target variables. For measuring vital capacity, the height of the patient is also given a linear scaling factor. Takamoto et al. 2020 [33] used multiple linear regression as well and included somatotype data in addition to height and BMI. Machine learning is used to infer scaling factors or directly to respiratory parameters based on extracted time, frequency, and subject-specific features (age, height, weight, gender). After feature selection, neural networks are used for regression [59].

### 3.8. Accuracy of the Measuring Methods

The accuracy of DPG methods is compared using gold-standard reference devices. For this purpose, a spirometer is operated in addition to the non-contact measurement, through which the patients breathe. If calibration of the model is needed, it can be realized in several ways, for example, intra-individually with reference measurement of the same subject or from a generalization of all subjects. When evaluating the results, a distinction can be made between:None: no transformation of the measured data was performed,Whole: the model was created with the whole data set,Subject: the model uses recordings of the same subject orMeasurement: the model uses test points of the same measurement.

Papers without a reference measurement device are not presented in this section. Only papers that report their deviations for tidal volume and vital capacity are shown in Table 1. For comparability, the results are given in mean ± standard deviation as far as possible.

Boudarham et al. 2013 [32] studied 20 subjects, including myotonic dystrophy type I (6 patients), diaphragmatic dysfunction (6), Pompe disease (5), spinal muscular atrophy (1), mitochondrial myopathy (1), and Duchennne muscular dystrophy (1). The determination of vital capacity with the OEP system shows a strong correlation (r² = 0.99, *p* < 0.001). The mean deviation is −20 mL, with limits of agreement at 163 mL and −203 mL. For all subjects, the deviation of the mean is <15%. The linear regression line is fitted with all data.

In Harte et al. 2016 [42], the quantification of the system with four Kinect cameras is carried out using a static evaluation and a dynamic evaluation. For a mannequin with a reference volume of 22.75 L, an RMS error of 0.100 L is determined. There is a significant correlation between the measurement signal with a spirometer signal (r² = 0.99, *p* < 0.001) for 22 patients. A comparison of respiratory parameters is not performed.

Oh et al. 2019 [49] present a system that does not require their own adaptation to the ten healthy subjects. The level set method produces a mean error of 8.41 ± 2.16% of the tidal volume, adjusted for two subjects with stronger outliers due to ventilation leak or body movements. The patients received the air volume through a ventilator during the measurements.

The determination of airway resistance by Ostadabass et al. 2016 [40] is performed using a Kincet camera to determine tidal volumes. There is a mean deviation of 0.07 ± 0.06 L of tidal volume. The 14 subjects were asked not to move during breathing.

The use of a smartphone by Reyes et al. 2017 [36] allows a root mean square error (RMSE) for the tidal volume of 0.182 ± 0.107 L. To calibrate the linear system, the data were trained using a training dataset of the same subject.

The evaluation according to Ripka et al. 2014 [21] shows a mean error of −30 mL in the lateral observation of the upper body by a camera. For this purpose, 50 healthy subjects were tested and a linear model was set up based on all data.

Sharp et al. 2017 [37] report the mean error as well as the percentage of measurements outside a range of ±150 mL on their system for forced vital capacity (19 mL, 0% outside ± 150 mL) forced expiratory volume in one second (82 mL, 61.9%), vital capacity (16 mL, 4.8%), and inspiratory capacity (23 mL, 6.0%). For this purpose, 100 subjects including COPD and asthma were tested with the single camera system. Using distinctive features from the signal, the volume-time signal is linearly regressively approximated. Such signal features include the extreme points of the measurement signal. Patients are encouraged not to move.

Soleimani et al. 2018 [43] compare a multiple-camera approach with a single-camera method. Mean and standard deviation are reported for forced vital capacity measurements and slow vital capacity measurements. Mean ± standard deviation is thus reported for vital capacity (−300 ± 561 mL), and tidal volume (0 ± 204 mL), among others, with the addition of one more camera, significantly reducing the error in each case. As in other measurements of Soleimani, the calibration of the system is done via leave-one-out cross-validation (LOOCV) using measurements of the same subject. Adding suppression of active body movements during breathing can further reduce the error [38]. However, the error values given are normalized and thus not directly comparable with other methods. The other datasets [48,57] of this working group restricted the active movement of the patient during the respiratory recordings, so that fewer errors occur [43].

The respiratory volume determination system by Takamoto et al. 2020 [33] requires patient height and BMI in real-world use. The goal of detecting COPD via VC and FEV1 is achieved with 81% sensitivity and 90% specificity. The errors in the determination of VC and FEV1 were repeated in a further measurement to check reproducibility.

Due to the sudden COVID-19 outbreak and subsequent restrictions, a study by Addison et al. 2021 [51] had to be stopped prematurely. Results with only one subject show a mean deviation of −213 ± 85 mL. Parallel results of respiration rate determination are significantly below with an RMSD of 0.5 breaths/min.

Two SROIs are combined with each other in Seppanen et al. 2015 [52]. Via least-squares FIR filter coefficients are determined in a test measurement, which processes the measurement signal of the SROIs non-linearly to the target signal. An absolute measurement error of 9.4 ± 8.4% is specified.

Transue et al. 2016 [60] propose a model based on iso-surface reconstruction. The chest motion is considered omni-directional and a 3D-model is created via surface-hole filling. Using a Bayesian neural network and the volume change of the 3D-model as input, the respiration volume is approximated. The deviations from the spirometer signal are given with a maximum of 7.8%. The accuracy is derived from a separate test data set. The training data set contains data from all four subjects.

A non-contact system, proposed by Imano et al. 2020 [55] was tested on 39 elderly people. A comparison was made between clothed and unclothed subjects, and men and women, with mean absolute relative errors ranging from 10.7% to 15.5%. After an individual, automatic ROI selection, tidal volume is determined via linear regression.

## 4. Discussion

### 4.1. ROI-Selection

When considering ROI, breathing mechanics play a critical role. The expansion of the thorax occurs in the vertical, transverse and sagital directions [61]. Thereby, the expansion in the anteroposterior direction is the largest and the contribution to the total change of the thoracic volume is the strongest.

In addition to thoracic movement, abdominal movement may be of greater importance, particularly in abdominal breathing. However, Kempfle et al. 2021 [41] show that the chest is the best region to extract respiratory signals and the abdominal region has the lowest signal quality for respiration rate determination with a depth camera. Differences in the interaction of the compartments may be due to different conditions. For example, differences were found in swimmers [26] and dancers [27] compared to untrained comparison subjects. Furthermore, there are different data on significant respiratory mechanics between the sexes. Some studies [62,63,64] found significant differences in thoracoabdominal movements, while other studies found no such differences [27,65]. Pathologic changes may also lead to changes in respiratory mechanics, as in spinal abnormalities [37] or partial lung defects [20], which may affect symmetry and patterns of breathing. A difference in lung function between the diseased and contralateral sides of the thorax after thoracotomy can be visualized by DPG as shown by Yu et al. 2012 [53].

Thus, it is shown that the subdivision of the chest into different subregions has diagnostic utility for respiratory assessment. Such subdivisions are obtained by marker positions or chest models that are manually placed in the image. However, for ease and speed of DPG-measurement, there is a need to perform automated selection and discrimination of subregions. This can be done using methods that are already used to determine the ROI itself. A subdivision into SROIs based on anatomy is possible, for example, by results of a skeletonization or body contour, as a first approach shown by [55].

### 4.2. Signal Reconstruction

Artifacts and disturbances that can couple into the measurement signal are caused by wrinkled clothing, additive overlapping movements, and uncovered areas that contribute to breathing. Wrinkled clothing creates additional volumes. For this reason, subjects are asked to completely avoid clothing on the upper body [28,42] or to use tight-fitting clothing [40]. Other wrinkles that may occur and lead to irregularities are eliminated by signal filtering, such as Butterworth filtering [66].

Trunk movements result from (1) a backward movement at the onset of deep inhalation, (2) a forward movement in the middle and after strong expiration [66]. The effects of motion artifacts are reported in [36,38,40,42,43,48,49,55]. The reduction of upper body motion can be counteracted by using multiple cameras, as shown by [38]. Thus, it is possible to view the upper body completely. A differentiation into involuntary movements is not necessary, the volume calculation is done with the entire upper body. Otherwise, measurements with strong motion artifacts can be removed manually [21] or via PCA [38]. Under the assumption that the shoulders do not interfere with the movement process of the upper body, a plane reference surface, through shoulders and hips, could be another method to reduce the disturbances due to movement. Such a surface could be determined, for example, by skeletal joints and greatly improve the signal quality in systems with only one camera.

As described in the previous section, different regions are also shown to have different contributions to respiratory mechanics. However, this is not addressed by using a single depth average over an ROI. Including additional SROIs can thus potentially further improve the measurement signal, especially if the concatenation of contributions from these compartments is non-linear–as indicated by the success of a PCA [38]. However, the assumption of almost all reviewed papers is a linear relationship between measured signal (volume of the upper body) with ground truth signal (volume of the lungs).

It can be seen that other personal characteristics correlate strongly with respiratory volumes, such as age [67], height [68] and weight [69], while a direct influence of gender is controversial and can be attributed to the before mentioned parameters [70]. Such demographic details seem to describe 50% of the variance factor of the scaling factor for linear regression [37]. These influencing factors can thus potentially also improve a DPG measurement [21,33,59].

Modeling the chest as mesh and total volume is contrasted with methods that use depth averages in the ROI as the measurement signal. Methods for 3D-model generation are much more computationally intensive, for example, the calculation of a respiratory cycle in Oh et al. 2019 takes about 10 minutes [49]. The averaging method is less computationally intensive and thus more suitable for potential real-world use [33]. It was shown that chest averaging is not inferior to 3D chest modelling [57]. To determine respiratory rate, different models were compared in Kempfle et al. 2021 [41]: a PCA-based model of uniformly distributed points in the ROI, mean and median from the ROI (raw), and mean and median from the ROI minus a reference area on the neck to suppress motion (diff), and a model [71] for detection and recovery of occluded regions (model). These different methods are shown in Figure 7. Their own model showed the best results regarding signal quality, but needs more computational time. In a sitting scenario without motion, non-diff-based methods show similar results as diff-based methods. For extracting the breathing signal, the median has been shown to be superior to using the mean. The examinations refer to normal breathing without deep breathing for the determination of vital capacity.

It turns out that the arbitrary movement of the upper body acts as the largest source of interference. While prohibiting movement is one approach, allowing free breathing and algorithmically eliminating movement interference would be beneficial for the real-world use of DPG. Especially when using only one camera, such filtering is made more difficult. Reference surfaces can take on a more apparent role in the future. At the same time, the use of non-linear models and multiple SROIs may also lead to a general improvement in the determination of respiration parameters. As shown, patient characteristics are related to the measured values.

### 4.3. Measurement Evaluation

To determine whether a system is applicable in practice or not, an error to a reference gauge is calculated. In the investigated studies, a spirometer always represented the ground truth signal, which can be adjusted for its time delay via cross-correlation with the measurement signal. Subsequently, an error value determination can be made via differences in the signal, or, as analysed in this review, via the respiratory parameters calculated from the signals. Spirometers themselves may have a maximum error of ± 2.5% according to the guidelines of the American Thoracic Society and European Respiratory Society [9]. However, such accuracy cannot be expected for a contactless system, as shown in the following.

The results of the methods in Section 3.8 already show acceptable results. However, the measurement situation and the recording conditions must be considered. Multiple camera systems and measurement methods, which must first be calibrated by a spirometer, do not correspond to the desired advantages of an uncomplicated, compact measurement. The same applies if the patient has to consciously suppress his movement, this no longer corresponds to the advantage of free breathing. There have been few satisfactory results in this area up to date.

In particular, calibration of a subject is necessary for most works. A linear model is mostly used for transforming the measured signal. This review was related to the measurement of respiratory volumes such as tidal volume and vital capacity, whereas other parameters such as FEV1 are equally of clinical interest. It can be seen that the simultaneous measurement of several such respiratory parameters is more challenging for a measurement system and yielding to greater errors. For an evaluation of a real application scenario, the model should not be created on the same data as it is being evaluated. Thus, the separation into training and test data is suitable, whereby at least the measurements of one person are separated from each other. If you want to test whether a calibration with a spirometer is necessary, Leave-One-Out-Cross-Validation should be used [72]. In this case, one subject corresponds to the test data set, while the model is created with the other subjects. The test subject alternates.

The goal of the DPG measurement method should be that it can be used calibration-free and without restriction in movement. When a system is used with new, unknown subjects, the same measurement results cannot be expected if the model was evaluated in advance using only known data. For this reason, future developments should consider withholding single subjects or performing LOOCV to evaluate their model and verify that the model can be applied calibration-free to new subjects.

## 5. Conclusions

Depth-based plethysmography (DPG) is a relatively new method, for determining respiratory parameters. As a potential mobile and low-cost system, DPG is suitable to measure respiratory parameters such as respiratory rate, tidal volume, and vital capacity without contact. Natural respiration is not affected and respiratory mechanics on the upper body can be visualized by DPG.

DPG can be divided into marker-based methods and direct depth estimation methods. Furthermore, it can be distinguished whether a single-camera system or a multiple-camera system is used. A common application scenario is in a standing, sitting, or lying position 1–2 m from the camera. In sitting and lying positions, upper body movements are suppressed, which superimposes the breathing movements. A restriction to tight-fitting clothing to reduce wrinkle impact is suggested. Depending on the procedure, a TOF or SL camera system should be selected.

The first step in measuring respiratory parameters is to select a region of interest. The selection of multiple regions is important in representing the contribution to the total volume and in evaluating diseases. The selection of an ROI can be based on skeletal joints or on body models, spatial or temporal information. Further processing of the measurement signals can be done using computationally intensive models, but in practical use, these are not superior to simple mean or median calculations. Especially for the filtering of superposed motion data, further signal processing steps are necessary. Possibilities are to use reference areas that are not affected by these movements or principle component decomposition and general monitoring with multiple cameras. In order to finally deduce the respiratory volume from the measured signal, a transformation of the measured signal is required. This is usually done using linear scaling functions. It has been shown that somatotype factors also have an influence. In the future, nonlinear models and machine learning can be used to approximate respiratory parameters using measurement signals from several regions and other patient-specific features. An evaluation of such models should be done using leave-one-subject-out cross-validation, so that the independence of the calibration of the model for the individual subject can be shown. Today’s systems show that they already achieve acceptable results with such a calibration or multiple cameras. However, to fully exploit the capabilities and advantages of DPG, future models must be designed as calibration-free, single-camera systems.

## Figures and Tables

**Figure 1 sensors-22-09680-f001:**
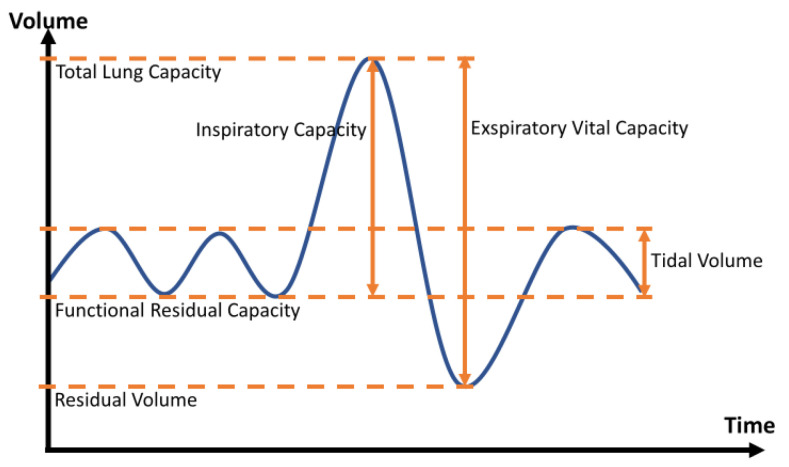
Schematic representation of the change in volumes over time.

**Figure 2 sensors-22-09680-f002:**
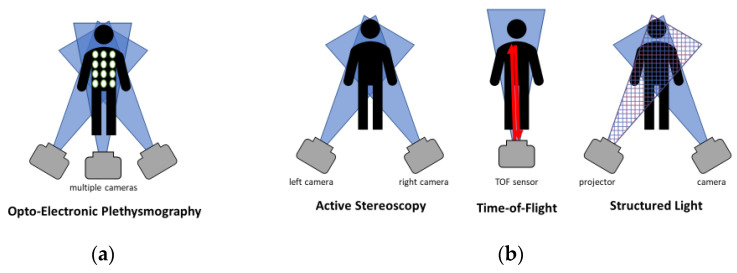
Marker-based and direct methods for measuring depth: (**a**) Use of markers and multiple cameras for depth determination by opto-electronic plethysmography; (**b**) Direct measurement methods to measure the depth values. Use of two offset cameras for active stereoscopy. Measurement of the path length of the reflected light in time-of-flight measurements. Measurement of the distortion of a projection pattern in structured light methods.

**Figure 3 sensors-22-09680-f003:**
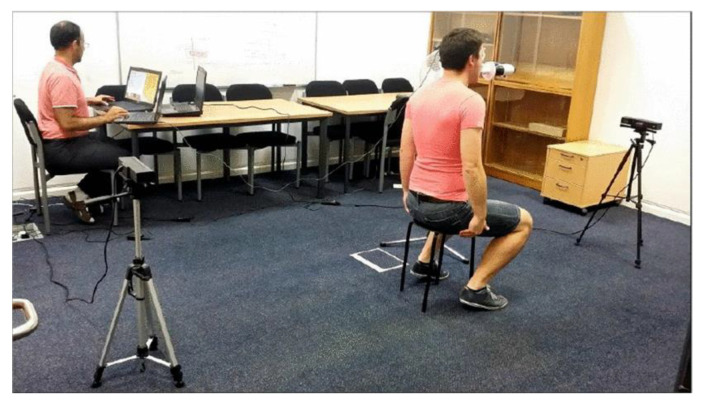
Measurement setup with two Kinect v2 cameras facing each other. The subject sits on a chair without a backrest and breathes through a reference spirometer. The image is taken from [43], licensed via Creative Commons License, and not changed for this work.

**Figure 4 sensors-22-09680-f004:**
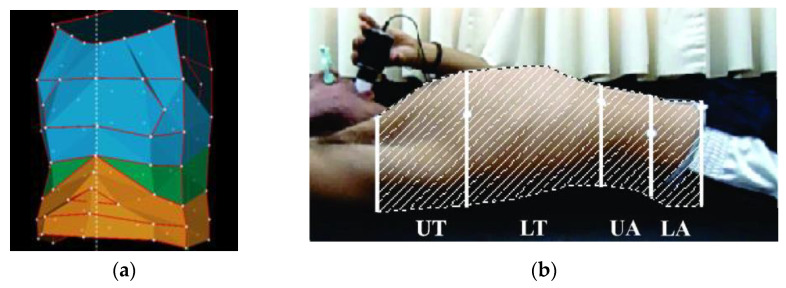
Selection of a region of interest for depth-based plethysmography: (**a**) Subdivision into three subregions: pulmonary rib cage (blue), abdominal rib cage (green), and the abdomen (orange). White dots represent attached markers. Image taken from [56], licensed via Creative Commons License and cropped for this work.; (**b**) Subregions in lateral view: upper thorax (UT), lower thorax (LW), upper abdomen (UA), lower abdomen (LA). Image taken from [21], licensed via Creative Commons Attribution License and cropped for this work.

**Figure 5 sensors-22-09680-f005:**
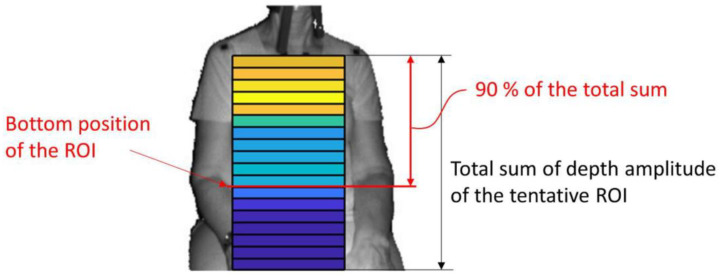
Determination of the final region of interest (ROI) as a selection of subregions. The areas are color coded with their corresponding amplitude strength from orange (strong) to blue (weak). Image taken from [55], licensed via Creative Commons Attribution License and not changed for this work.

**Figure 6 sensors-22-09680-f006:**
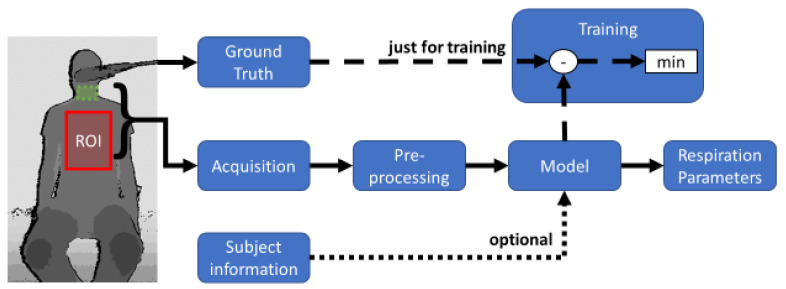
Flow chart for non-contact determination of respiratory parameters. On the left, a depth image with a region of interest (ROI, red) on the chest and subROI on the neck (green). The acquired data is preprocessed and then transformed into a model. Optionally, patient data can be added to the model. In a training session, the error is minimized with ground truth.

**Figure 7 sensors-22-09680-f007:**
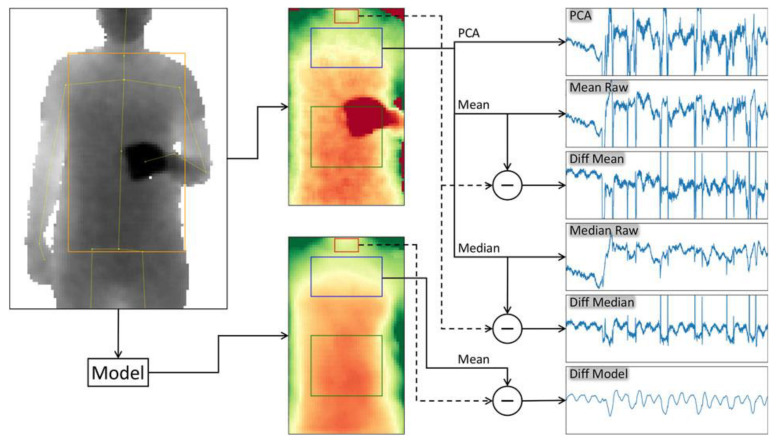
Comparison of six different methods for determining respiration rate. Principal component analysis (PCA) from the raw signal of the region of interest (ROI). Mean of ROI (Mean Raw), the median of ROI (Median Raw). Use of a reference plane to subtract motion about the mean (Diff Mean) and median (Diff Median). Application of an own model with the help of a reference plane (Diff Model). Figure taken from [41], licensed via Creative Commons Attribution License and not changed for this work.

**Table 1 sensors-22-09680-t001:** Comparison of results for measuring tidal volume (TV) and vital capacity (VC).

Paper	Marker/ Direct	Cameras	Model	Calibration	Results for Respiration Parameters	#Subjects (Healthy)
[32]	marker	multiple	linear	whole	VC: mean: −20 ± 93 mL	20 (0)
[49]	direct	single	linear	none	TV: mean: 8.41%	10 (10)
[40]	direct	single	linear	none	TV: mean: 70 ± 60 mL	14 (14)
[36]	direct	single	linear	subject	TV: RMSE 182 ± 107 mL	15 (15)
[21]	marker	single	linear	whole	VC: mean: −30 ± 352 mL	50 (50)
[37]	direct	single	linear	whole	VC: mean: 16 ± 51 mL	100 (21)
[43]	direct	multiple	linear	subject	VC: mean: −300 ± 561 mL TV: mean: 0 ± 204 mL	35 (35)
[57]	direct	single	linear	subject	VC: mean: −150 ± 842 mL TV: mean: 100 ± 255 mL	35 (35)
[48]	direct	single	linear	meas.	VC: mean: 9 ± 39 mL TV: mean: 74 ± 88 mL	40 (0)
[33]	direct	single	linear	none	VC: mean: 57 ± 716 mL	53 (21)
[51]	direct	single	linear	subject	TV: mean: −213 ± 85 mL	1 (1)
[52]	direct	single	non-linear	subject	TV: mean: 9.4 ± 8.4%	8 (8)
[60]	direct	single	non-linear	subject	TV: max: 7.8%, min: 5.81%	4 (4)
[55]	direct	single	linear	whole	TV: mean: 10.7% up to 15.5%	39 (39)

## Data Availability

Not applicable.

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
