# Peer review of "Depth-Based Measurement of Respiratory Volumes: A Review"

_sensors, 2022, doi:10.3390/s22249680_

Round 1

Reviewer 1 Report

Review paper that provides an overview of technical advances in the estimation of respiratory volumes using depth-based measurements. The manuscript offers a broad discussion on technical aspects and the advantages with respect to wearable devices. The review begins by presenting the concepts involved in obtaining information using depth sensing cameras, and then explains their use in monitoring information on respiratory volumes, specifically tidal volume and vital capacity. In general, the manuscript is clear, well written and organized, but I have some comments that, I hope, will contribute to increasing the understanding of the subject and the quality of the manuscript.

 Major comments:

Given the recent publication of review papers (2020 and 2021) on depth-based respiratory measurements, what would be the advantages of the new revision proposed for 2022? I suggest extending, in the Introduction section, some comments on these previous review works, pointing out the differences and contributions of the current review.

 References are adequate and I understand that the manuscript focuses on measurements of tidal volume and vital capacity. However, during 2020 and 2022 the interest in contactless measurements has increased, so some papers have been published (most of them in Sensors) that could be related to the subject of the current manuscript. For example:

1)      Imano, W.; Kameyama, K.; Hollingdal, M.; Refsgaard, J.; Larsen, K.; Topp, C.; Kronborg, S.H.; Gade, J.D.; Dinesen, B. Non-Contact Respiratory Measurement Using a Depth Camera for Elderly People. Sensors (Basel). 2020, 20, 6901. https://doi.org/10.3390/s20236901

2)      Romano C, Schena E, Silvestri S, Massaroni C. Non-Contact Respiratory Monitoring Using an RGB Camera for Real-World Applications. Sensors (Basel). 2021 Jul 29;21(15):5126. doi: 10.3390/s21155126.

3)      Yang F, He S, Sadanand S, Yusuf A, Bolic M. Contactless Measurement of Vital Signs Using Thermal and RGB Cameras: A Study of COVID 19-Related Health Monitoring. Sensors (Basel). 2022 Jan 14;22(2):627. doi: 10.3390/s22020627.

4)      Molinaro N, Schena E, Silvestri S, Massaroni C. Multi-ROI Spectral Approach for the Continuous Remote Cardio-Respiratory Monitoring from Mobile Device Built-In Cameras. Sensors (Basel). 2022 Mar 25;22(7):2539. doi: 10.3390/s22072539.

5)      Selvaraju V, Spicher N, Wang J, Ganapathy N, Warnecke JM, Leonhardt S, Swaminathan R, Deserno TM. Continuous Monitoring of Vital Signs Using Cameras: A Systematic Review. Sensors (Basel). 2022 May 28;22(11):4097. doi: 10.3390/s22114097.

(Note: Please check citation format of these references)

 Minor comments:

In the text, depth-based measurement is proposed as an alternative to spirometry and body plethysmography. However, it is not specified in the text that these techniques, and particularly plethysmography, allow testing not only volumes, but also a wide variety of respiratory mechanics parameters, including total lung capacity, airway resistance, lung and thoracic compliance, among other. Could you add a comment about it?

At the end of the Introduction there is a brief description of the content of each section of the manuscript. However, at the beginning of each section, a description of their content is made again, which seems excessive and unnecessary.

 Other specific commnents;

Line 57. Change “work” to “review”

Line 62. Change “parameters” to “volumes”

Line 64. Reference [8] is not suitable for the accompanying text.

Line 161. Change “trachestomy” to “tracheostomy”

Line 271. Change “umbilicius” to “umbilicus”

Line 271. Define acronym PCA as it appears for the first time. Then, avoid definition in line 333.

Line 294, Figure 5. Write Akquisition in English.

Line 365. Change “sponal” to “spinal”

Line 500. Change “principle” to “principal”

Line 713. Correct citation: “Addison PS, Smit P, Jacquel D, Addison AP, Miller C, Kimm G. Continuous non-contact respiratory rate and tidal volume monitoring using a Depth Sensing Camera. J Clin Monitor Comput. 2022, 36(3), 657-665. doi: 10.1007/s10877-021-00691-3.”

Author Response

Dear Reviewer,

Thank you for your time and constructive feedback.

In the attachment, we address the points you suggested.

Reviewer 2 Report

1.      In this study, method was defined for Depth-based plethysmography (DPG) and a limited number of articles were reviewed. Actually, I didn't realize that it was a developmental process as mentioned in the title.

2.      Reference lists and discussion sections may be not so long

3.      If the authors have more pictures of Depth-based plethysmography (DPG) can be added to better explain the ROIs

Author Response

Dear Reviewer,

Thank you for your time and constructive feedback.

In the following, we address the points you suggested:

  1. In this study, method was defined for Depth-based plethysmography (DPG) and a limited number of articles were reviewed.
    The limited number of included papers results from the chosen focus of this review: Volume measurement using DPG. At the suggestion of another reviewer, an additional paper was added.
    Actually, I didn't realize that it was a developmental process as mentioned in the title.
    We changed the title to: "Depth-based Measurement of Respiratory Volumes: A Review"

  2. Reference lists and discussion sections may be not so long 
    We see a great added value in our work not only as a summary of existing literature, but also in the discussion of that literature to address the strengths and weaknesses of each method and to guide future developments. For this reason, it is also necessary to include a sufficient number of sources from the literature and to discuss the respective different approaches. We have therefore decided not to shorten the discussion and the bibliography.

  3. If the authors have more pictures of Depth-based plethysmography (DPG) can be added to better explain the ROIs
    Another work with corresponding presentation of the selected ROI was include, see figure 5 in section 3.6.

Reviewer 3 Report

This is a comprehensive, well researched, and very timely narrative review. There is currently a lack of synthesis in the research community with regards to the use of DPG to measure respiratory rate and I suspect that this review will be used as a reference to those with interest in the field, especially as DPG becomes more frequently used as a non contact modality. 

Author Response

Dear Reviewer,

we would like to thank you for your time.
Thank you for your support and encouragement of our work.

There were no suggestions for improvement to be included. 

Round 2

Reviewer 1 Report

I consider that the authors have taken into account all the comments made to the first version, so I am satisfied with the answers of the authors and the modifications included in the new version.

One last minor comment:

Although the new version has enough clarity, quality and originality to be accepted for publication as a review article, the new Figure 5 should be checked carefully because the text do not match the colors.

Author Response

Again, we would like to thank the reviewer for taking the time to thoroughly review the revisions. 

Although the new version has enough clarity, quality and originality to be accepted for publication as a review article, the new Figure 5 should be checked carefully because the text do not match the colors.

We have adjusted the Figure 5 caption to match the illustration.